# Impact of the Genotype and Phenotype of CYP3A and P-gp on the Apixaban and Rivaroxaban Exposure in a Real-World Setting

**DOI:** 10.3390/jpm12040526

**Published:** 2022-03-24

**Authors:** Camille Lenoir, Jean Terrier, Yvonne Gloor, Pauline Gosselin, Youssef Daali, Christophe Combescure, Jules Alexandre Desmeules, Caroline Flora Samer, Jean-Luc Reny, Victoria Rollason

**Affiliations:** 1Department of Anaesthesiology, Pharmacology, Intensive Care and Emergency Medicine, Division of Clinical Pharmacology and Toxicology, Geneva University Hospitals, 1205 Geneva, Switzerland; camille.lenoir@hcuge.ch (C.L.); jean.terrier@hcuge.ch (J.T.); yvonne.gloor@hcuge.ch (Y.G.); youssef.daali@hcuge.ch (Y.D.); jules.desmeules@hcuge.ch (J.A.D.); caroline.samer@hcuge.ch (C.F.S.); 2Institute of Pharmaceutical Sciences of Western Switzerland, University of Geneva, 1206 Geneva, Switzerland; 3Department of Medicine, Division of General Internal Medicine, Geneva University Hospitals, 1205 Geneva, Switzerland; pauline.gosselin@hcuge.ch (P.G.); jean-luc.reny@hcuge.ch (J.-L.R.); 4Geneva Platelet Group, Faculty of Medicine, University of Geneva, 1206 Geneva, Switzerland; 5Faculty of Medicine, University of Geneva, 1206 Geneva, Switzerland; christophe.combescure@hcuge.ch; 6Department of Health and Community Medicine, Division of Clinical Epidemiology, Geneva University Hospitals, 1205 Geneva, Switzerland

**Keywords:** DOACs, pharmacogenomics, phenotype, metabolism, personalized medicine

## Abstract

Apixaban and rivaroxaban are the two most prescribed direct factor Xa inhibitors. With the increased use of DOACs in real-world settings, safety and efficacy concerns have emerged, particularly regarding their concomitant use with other drugs. Increasing evidence highlights drug–drug interactions with CYP3A/P-gp modulators leading to adverse events. However, current recommendations for dose adjustment do not consider CYP3A/P-gp genotype and phenotype. We aimed to determine their impact on apixaban and rivaroxaban blood exposure. Three-hundred hospitalized patients were included. CYP3A and P-gp phenotypic activities were assessed by the metabolic ratio of midazolam and AUC_0–6h_ of fexofenadine, respectively. Relevant *CYP3A* and *ABCB1* genetic polymorphisms were also tested. Capillary blood samples collected at four time-points after apixaban or rivaroxaban administration allowed the calculation of pharmacokinetic parameters. According to the developed multivariable linear regression models, P-gp activity (*p* < 0.001) and creatinine clearance (CrCl) (*p* = 0.01) significantly affected apixaban AUC_0–6h_. P-gp activity (*p* < 0.001) also significantly impacted rivaroxaban AUC_0–6h_. The phenotypic switch (from normal to poor metabolizer) of P-gp led to an increase of apixaban and rivaroxaban AUC_0–6h_ by 16% and 25%, respectively, equivalent to a decrease of 38 mL/min in CrCl according to the apixaban model. CYP3A phenotype and tested SNPs of CYP3A/P-gp had no significant impact. In conclusion, P-gp phenotypic activity, rather than known CYP3A/P-gp polymorphisms, could be relevant for dose adjustment.

## 1. Introduction

Apixaban and rivaroxaban are the two most prescribed direct oral anticoagulants (DOACs), both acting by direct inhibition of factor Xa (FXa) [1]. DOACs have become the treatment of choice for the treatment and prophylaxis of deep vein thrombosis (DVT) and pulmonary embolism (PE), as well as for the reduction of the risk of stroke and embolism in non-valvular atrial fibrillation (AF) [2,3,4,5]. Guidelines shifted from vitamin K antagonists (VKA) to DOACs, with DOACs being promoted as having a lower propensity to interact with drugs and food, a better predictable anticoagulant effect, and the ability to be administered in fixed doses without routine monitoring [3,6]. Dose adjustment of DOACs is nonetheless required in specific risk groups [7]. For instance, dosing depends on indication, age (>80 years), body weight (<60 kg), and serum creatinine level (>1.5 mg/dL) for apixaban [2,8]. For rivaroxaban, dosing depends on indication and creatinine clearance (CrCl) values (CrCl < 50 mL/min) [2,9]. However, effectiveness and safety concerns in addition to significant inter-individual variations in dose–concentration response have been observed following their use in real-world settings, outside the strictly monitored conditions of the clinical trials [7,10].

Although apixaban and rivaroxaban are substrates of cytochrome P450 (CYP) 3A4/5 (3A) and P-glycoprotein (P-gp), variables impacting CYP3A and P-gp activity or expression (e.g., drug–drug interactions (DDIs) and genotypes) are not considered for dose adaptation [11]. This is a caveat, as the activity of drug transporters and metabolizing enzymes can be inhibited or induced by genetic, environmental, physiological, and pathophysiological factors, leading to DOACs’ under- or overexposure [12]. The concern is important, as the risk of adverse drug reactions (ADRs) such as bleeding or thromboembolism increases with the occurrence of out-of-target concentrations [13]. It is also enhanced with polymedication, with a study showing that 30% of patients treated by DOACs received at least one interacting drug [7,14,15]. The increased bleeding risk due to coadministration with CYP3A and P-gp inhibitors is more and more reported in the literature, through case reports and several large registry-based retrospective studies [16,17,18,19,20,21]. The occurrence of thromboembolic events is also described in the literature after the concomitant use of apixaban and rivaroxaban with CYP3A and P-gp inducers [19,20]. Summaries of Product Characteristics (SmPC) only suggest avoiding concomitant use with strong CYP3A and P-gp inhibitors, without a benefit/risk evaluation [14,18]. There are currently no guidelines concerning coadministration with moderate or mild modulators, as data are rare and conflicting [14]. Numerous ADRs following DDIs involving apixaban or rivaroxaban and CYP3A/P-gp modulators were identified in VigiBase, the World Health Organization (WHO) database [19,20]. However, data on clinically relevant ADRs with DOACs due to DDIs and specific plasmatic concentrations inducing ADRs are scarce [22].

In addition to DDIs, the observed inter-individual variability in apixaban and rivaroxaban exposure could be related to polymorphisms of genes coding for CYP3A and/or P-gp, as recently reviewed [23]. Indeed, the effectiveness and safety of DOACs are influenced by genetically determined characteristics involved in drug metabolism [24]. For instance, a study found a significant association between the intronic variant *rs4148738* of *ABCB1* gene, coding for P-gp, and an increase in the peak concentration of apixaban [25]. Studies found that the presence of homozygous mutated TT genotype for *rs2032582* and *rs1045642* induced bleeding during rivaroxaban treatment [26]. However, other studies found that these variants had no significant impact [27]. The variants 1236C>T (*rs1128503*), 2677G>T (*rs2032582*), and 3435C>T (*rs1045642*) of the *ABCB1* gene had no impact on the concentration/dose ratio of apixaban [28]. Regarding CYP3A, the presence of CYP3A5*1/*3 or *3/*3 diplotypes was associated with an increase of apixaban concentration/dose ratio, compared to CYP3A5*1/*1 [28,29]. Nevertheless, conflicting results were reported, as a study found no significant impact of the CYP3A5*3 genetic polymorphism [30]. In addition, a study found that CYP3A4 activity had an impact on the peak and trough concentrations of rivaroxaban, while diverging results also exist [31,32].

Overall, selecting the suitable dose of DOACs is a complex process with different criteria and factors involved [33]. Data suggest a significant impact of factors altering CYP3A/P-gp activity, such as gene polymorphisms and DDIs, on rivaroxaban and apixaban exposure. In order to study the overall effect of such genetic and environmental effects on DOACs exposure, we used a validated cocktail approach with specific exogenous probes to prospectively determine whether CYP3A/P-gp phenotypic activities had a significant impact on apixaban and rivaroxaban exposure in hospitalized patients [7,13]. This real-life setting allowed us to ensure the establishment of a cohort displaying a large inter-individual variability in CYP3A/P-gp phenotypic activities caused by a broad panel of covariables. This facilitated the study of their impact on DOAC exposure. The second aim of this study was to assess the impact of relevant gene polymorphisms for CYP3A/P-gp encoding genes on drug exposure.

## 2. Materials and Methods

### 2.1. Study Design

This study, investigating the impact of CYP3A and P-gp genotype and phenotype on blood concentrations of apixaban and rivaroxaban, was a real-life prospective observational study. The study protocol was registered on the US National Institutes of Health clinical trials registry (NCT03112525) and approved by the regional research ethics committee of the canton of Geneva (CCER) (No. 2016-01490, date of approval: 25 January 2017). Written informed consent was obtained from all participants prior to the initiation of any study procedure. The study complied with the principles of the Declaration of Helsinki and International Conference on Harmonization Good Clinical Practice Guidelines.

### 2.2. Study Population

Patients were recruited during their hospitalization at the Geneva University Hospitals between June 2017 and January 2021. Eligible patients were 18 years or above, diagnosed with AF, DVT, PE, and treated with apixaban or rivaroxaban for at least three days at the same dosage to ensure steady-state. Exclusion criteria included any known allergy to one of the components of the “Geneva cocktail” (caffeine, bupropion, flurbiprofen, omeprazole, dextromethorphan, midazolam, and fexofenadine). Patients were selected based on their electronic health record after a prescription alert was received for apixaban or rivaroxaban. Comedications were systematically screened to record patients taking CYP3A4/5 and/or P-gp inhibitors and/or inducers using the Lexicomp drug interaction analysis tool and the Geneva Table of CYP substrates, inhibitors, and inducers [34,35,36]. Adequacy of dosage was assessed according to the SmPC criteria (indication, age, CrCl, and weight).

### 2.3. Genotyping of CYP3A4/5 and P-Glycoprotein Encoding Genes

DNA was isolated from whole blood anticoagulated with EDTA, collected from each study participant prior to or following phenotype blood sampling, with a QIAsymphony^®^ SP/AS (QIAGEN, Hilde, Germany) instrument using the QIAsymphony^®^ DSP DNA Midi Kit (QIAGEN, Hilden, Germany) according to the manufacturer’s protocol. Qubit^TM^ fluorometer (ThermoFisher Scientific, Life Technologies Holdings Pte Ltd., Singapore) was used afterwards to quantify the purified DNA and ensure that the samples were at the normalized concentration of 30 ng/μL.

Genotyping of selected *CYP3A4/5* and *ABCB1* polymorphisms was carried out on QuantStudio^TM^ 12K Flex Real-Time (RT) PCR System with TaqMan^®^ OpenArray^TM^ genotyping assays and TaqMan^®^ MGB Probe Validated Single Nucleotide Polymorphism (SNP) Genotyping Assays (Applied Biosystems, Thermo Fisher Scientific, Waltham, MA, USA), respectively. These methods were previously described in detail in the literature [37,38,39]. In our study, SNPs rs1045642 (3435C>T), rs1128503 (1236C>T), and rs2032582 (2677G>T/A) of *ABCB1* were investigated. All SNPs of CYP3A4/5 integrated in the TaqMan^®^ OpenArray^TM^ PGx Express Panel (Thermo Fisher Scientific, Waltham, MA, USA) were considered and are listed in Appendix A. Finally, raw genotyping data were processed with the TaqMan^®^ Genotyper software (Thermo Fisher Scientific, Waltham, MA, USA) according to the manufacturer’s protocol.

AlleleTyper™ Software and translational tables (Thermo Fisher Scientific and PharmGKB, Stanford, CA, USA) were used to translate genetic pattern information from genotyping (SNPs) to pharmacogenomic gene-level star (*) nomenclature. Allele and genotype frequencies were determined, and Hardy–Weinberg equilibrium (HWE) was tested. The HWE is respected if the chi-squared (χ^2^) value is less than 3.84, because 3.84 is the threshold value for a significance level of *p* = 0.05 for one degree of freedom. Samples with call rates below 95% were excluded from analysis. The CYP3A activity predicted from the genotype combines the effects of CYP3A4 and CYP3A5 SNPs on enzyme activities, as listed in the PharmVar and PharmGKB databases [40,41]. Patients were classified into poor metabolizer (PM), intermediate metabolizer (IM), normal metabolizer (NM), and ultra-rapid metabolizer (UM) according to the classification described by Andreu et al. [42].

### 2.4. Phenotyping

The phenotypic activity of CYP3A4/5 and P-gp was measured by calculating the metabolic ratio (MR) and the area under the curve (AUC_0–6h_) of the probe substrates, respectively. The “Geneva cocktail” (caffeine 50 mg, CYP1A2; bupropion 20 mg, CYP2B6; flurbiprofen 10 mg, CYP2C9; omeprazole 10 mg, CYP2C19; dextromethorphan 10 mg, CYP2D6; midazolam 1 mg, CYP3A; and fexofenadine 25 mg, P-gp) was administered orally on an empty stomach. Capillary blood samples were collected two (t + 2 h), three (t + 3 h), and six (t + 6 h) hours later with dried blood spots (DBS), using a previously validated sampling method, and were stored at −20 °C in a sealable plastic bag until analysis [43,44]. MR of CYP3A4/5 consists of the blood concentration of 1-OH-midazolam divided by the blood concentration of midazolam measured after two hours. The activity of P-gp is derived from the AUC_0–6h_ of fexofenadine (AUC_fexofenadine_), calculated by linear trapezoidal rule using WinNonlin^®^ version 6.2.1 (Pharsight, Mountain View, CA, USA) from blood concentrations of fexofenadine measured at t + 2 h, t + 3 h, and t + 6 h. A previously validated method using a liquid chromatography-tandem mass spectrometry (LC-MS/MS) quantification was used to assess the blood concentrations of midazolam, 1-OH-midazolam, and fexofenadine [43,45,46].

### 2.5. Laboratory Markers Levels

Whole-blood samples with lithium heparin were collected early in the morning on the study day to assess liver and renal function. The concentration of aspartate transaminase (ASAT), alanine transaminase (ALAT), alkaline phosphatase (ALP), gamma-glutamyltransferase (GGT), bilirubin, and creatinine were measured directly after blood sampling. The CrCl was calculated according to the Cockcroft–Gault formula and used as a continuous variable in our multivariable linear regression models. However, to describe the population, patients were classified based on their CrCl into normal (>60 mL/min/1.73 m^2^), moderate (30 < x < 59 mL/min/1.73 m^2^), severe (15 < x < 29 mL/min/1.73 m^2^), and end-stage renal disease (<15 mL/min/1.73 m^2^). Patients were also classified into normal or abnormal liver function (defined as ASAT, ALAT, bilirubin, GGT > 2 × upper limit of normal).

### 2.6. Apixaban and Rivaroxaban Blood Concentrations

Capillary blood samples (10 µL) were collected in DBS just before apixaban and rivaroxaban administration (t0) and at t + 2 h, t + 3 h, and t + 6 h. The DBS concentrations were determined using a validated LC-MS/MS method. The instrumentation used was composed of an Agilent 1290 Infinity series LC system from Agilent (Paolo Alto, Santa Clara, CA, USA) coupled to a 6500 QTtrap^®^ triple quadrupole linear ion trap mass spectrometer from AB Sciex equipped with an electrospray ionization (Darmstadt, Germany).

Before analysis, discs (i.d. 8 mm) covering the entire DBS were punched out, placed in LC vials, and extracted by adding 100 μL of methanol containing 200 ng/mL of internal standards (apixaban-d3 and rivaroxaban-d4). After agitation during 10 min, 10 μL of the supernatant was injected into the LC-MS/MS system. Separation was performed with a Kinetex^®^ C18 column (50 × 2.1 mm, 2.6 µm) from Phenomenex (Brechbühler, Schlieren, Switzerland) under gradient conditions. The mobile phase was composed of formic acid 0.1% in water and in acetonitrile. The total run time was 7 min. Detection of analytes was obtained in positive mode using multiple reaction monitoring (MRM). Instrument parameters were as follows: curtain gas = 40 psi, collision gas = high, IonSpray voltage = 4500 kV, temperature = 550 °C, ion source gas 1 = 60 psi, ion source gas 2 = 60 psi. The transitions monitored for each analyte (precursor ion > product-fragment ions) were: apixaban 460.1 > 443.1, apixaban-d3 463.1 > 446.1, rivaroxaban 436.0 > 144.8, and rivaroxaban-d4 440.1 > 144.8. The optimized collision energy was +33 V for apixaban and +35 V for rivaroxaban. Declustering potential (DP) was +156 V for apixaban and +136 V for rivaroxaban. Cell exit potential (CXP) was +36 V for apixaban and +16 V for rivaroxaban.

The calibration curves were linear over the standard concentration ranges of 1–1000 ng/mL for all analytes and trueness; inter and intraday variabilities were in line with the validation guidelines of the European Medicines Agency. The AUC_0–6h_ of apixaban and rivaroxaban were calculated by linear trapezoidal rule using WinNonlin^®^ version 6.2.1 (Pharsight, Mountain View, CA, USA).

### 2.7. Statistical Analysis

A sample size of 204 patients treated with apixaban was needed to detect a difference of at least 50 ng/mL in mean plasma concentration of apixaban between patients with high or low enzymatic activity (CYP3A4/5 and P-gp) with a power of 80% and a two-sided α-value of 5%. A standard deviation (SD) of 100 ng/mL in each group was assumed.

Concerning rivaroxaban, a sample size of 150 patients treated with rivaroxaban was needed to detect a difference of at least 60 ng/mL in mean plasma concentration of rivaroxaban between patients with high or low enzymatic activity (CYP3A4/5 and P-gp) with a power of 80% and a two-sided α-value of 5%. A SD of 100 ng/mL in each group was assumed. For both molecules, a normal CYP3A4/5 activity, as predicted by the genotype, was expected in 20% of enrolled patients. Indeed, the CYP3A5*3 mutation has a reduced activity and is highly prevalent in the population worldwide, especially in Caucasians [47].

Dependent variables (outcomes) for both drugs were defined as being the values of AUC_0–6h_ and concentration 2 h after drug administration (C_2h_). All statistical analyses were performed using the software R version 4.0.2 (R Core Team, 2016. R: A Language and Environment for Statistical Computing, Vienna, Austria), and a *p*-value < 0.05 was considered statistically significant. Means ± SD were used to describe continuous variables. Associations of variables with outcomes were investigated with multivariable linear regression models. Each independent variable is reported with its beta coefficient (β) and its 95% confidence interval (CI95%). For continuous independent variables (MR_midazolam_, AUC_fexofenadine_, BMI, CrCl, and age), the linearity of the relationship was graphically inspected. Because of the skewness of the distribution of MR_midazolam_ and AUC_fexofenadine_, a log_10_ transformation was applied. The Breusch–Pagan test was used to detect a potential heteroscedasticity issue, and consistent standard errors of the regression coefficients were assessed with a sandwich estimator. Association between dependent variables and phenotypic activity of CYP3A and P-gp (MR_midazolam_ and AUC_fexofenadine_, respectively) were adjusted for a pre-specified set of potential confounding factors (gender, CrCl, BMI, age, dose). In addition, association between dependent variables and predicted activity of CYP3A and P-gp from genotype were also adjusted for a pre-specified set of potential confounding factors (gender, CrCl, BMI, age, dose). Spearman’s correlation was used to assess the concordance between genotype and phenotype of CYP3A and P-gp. Missing data were excluded from the analysis.

## 3. Results

### 3.1. Demographics

Overall, 300 patients were included, with 164 receiving apixaban and 136 receiving rivaroxaban. Although lower than anticipated, sample sizes allowed detection of a difference of at least 57 ng/mL and 62 ng/mL in mean plasma concentration of apixaban and rivaroxaban, respectively, with a power of 80% and a two-sided α-value of 5%. The only patient receiving rivaroxaban 2.5 mg twice daily (bid), a new indication in association with aspirin for the prevention of atherothrombotic events in high-risk patients, was removed from the analysis. Demographic characteristics of the study population are presented in Table 1.

### 3.2. Genotypes

Of the 299 remaining patients, 294 patients were successfully genotyped. Frequencies for each SNP studied are presented in Table 2. Five patients were not genotyped due to missing blood sampling. Predicting P-gp phenotype from genotype was impossible because of the lack of clear haplotype–phenotype correlations. The final sample and SNP call rates for the whole analysis (CYP3A4, CYP3A5, and ABCB1) was 99.6% and 99.4%, respectively. No significant departure from HWE was found for all the SNPs, except for CYP3A4*1B (χ^2^ = 11.25, *p* = 0.001). The frequencies of the different genotypes were relatively similar between the two drugs studied and correspond to the reference population (Caucasian) [47]. Data are shown in Appendix A.

### 3.3. Phenotypes

All the included patients received the Geneva cocktail, but the fexofenadine AUC_0–6h_ calculation is missing for one patient who received rivaroxaban treatment, because the sampling at t + 3 h and t + 6 h was not possible. The means ± SD of MR_midazolam_ in the apixaban and rivaroxaban cohorts are 0.62 ± 0.67 and 0.58 ± 0.58, respectively. The means ± SD of AUC_fexofenadine_ in the apixaban and rivaroxaban cohorts are 265.0 ± 178.0 ng × h/mL and 237.9 ± 170.0 ng × h/mL, respectively. Results are summarized in Figure 1; Figure 2 for apixaban and rivaroxaban, respectively. The calculated inter-individual variability (coefficient of variation (CV)) of MR_midazolam_ and AUC_fexofenadine_ for the apixaban cohort is of 108.1% and 67.2%, respectively. The CV of MR_midazolam_ and AUC_fexofenadine_ for the rivaroxaban cohort is of 100.0% and 71.5%, respectively. Spearman‘s correlation coefficient between MR_midazolam_ and AUC_fexofenadine_ is ρ = −0.271 (*p* < 0.0001).

### 3.4. Apixaban and Rivaroxaban Blood Concentrations

Individual pharmacokinetic (PK) profiles for patients treated with apixaban bid, rivaroxaban once daily (od), and rivaroxaban bid are presented in Figure 3a–c, respectively. The corresponding mean ± SD is highlighted in red. All blood concentrations were normalized by the dosing regimen. The inter-individual CV of blood concentrations is 47.7% for apixaban bid, 51.8% for rivaroxaban administered od, and 41.5% for rivaroxaban administered bid.

Spearman’s correlation coefficient between blood concentrations at t + 2 h and t + 3 h were ρ = 0.94 (*p* < 0.0001) and ρ = 0.82 (*p* < 0.0001) for apixaban and rivaroxaban, respectively. As we observed a good correlation, the C_2h_ was used for the analysis.

### 3.5. Multivariable Linear Regression

A multivariable linear regression model was built to assess the factors associated with the AUC_0–6h_ or C_2h_ of apixaban and rivaroxaban. The models built for the AUC_0–6h_ of apixaban and rivaroxaban are shown in Table 3 and predict 47% and 27% of the observed variability, respectively.

The models built for the C_2h_ of apixaban and rivaroxaban are shown in Appendix A.

After adjustment for all the covariables, the P-gp activity and the dose administered have a positive and significant association with AUC_0–6h_ and C_2h_ of apixaban and rivaroxaban. In addition, the CrCl is negatively and significantly associated to AUC_0–6h_ and C_2h_ of apixaban, while this is not the case with rivaroxaban. In practice, an increase in fexofenadine AUC_fexofenadine_ from 100.1 ng × h/mL to 285.5 ng × h/mL (corresponding to a phenotype conversion from NM to PM according to our inner threshold values) would lead to an increase in apixaban and rivaroxaban AUC_0– 6h_ by about 16% and 25%, respectively [43,44]. For apixaban, this P-gp phenoconversion can be compared to the effect of a decrease in CrCl of 37.6 mL/min/1.73 m^2^. For rivaroxaban, this phenoconversion is equivalent to an increase in the dose category (see Table 3).

Age, gender, weight, ALAT level, and CYP3A activity were not associated with AUC_0–6h_ and C_2h_ variations of apixaban and rivaroxaban.

The same multivariable linear regression models were built to assess the impact of activity predicted from genotype of CYP3A and P-gp with the PK parameters of apixaban and rivaroxaban. Models built for the AUC_0–6h_ of apixaban and rivaroxaban are shown in Table 4 and explain approximately 40% and 18% of the observed variability, respectively.

The models built for the C_2h_ of apixaban and rivaroxaban are shown in Appendix A.

No SNP of P-gp tested was found to be associated with apixaban and rivaroxaban AUC_0−6h_ (Table 4) or C_2h_ (Appendix A). The CYP3A predicted activity from genotype was found to be associated with AUC_0−6h_ (Table 4) or C_2h_ (Appendix A) of rivaroxaban but not apixaban. However, these results showed that being NM paradoxically increases the exposure to rivaroxaban, as compared to IM (Appendix A). It highlights the difficulty to predict the CYP3A phenotype from genotype, which is confirmed by the absence of correlation between CYP3A predicted by genotype and MR_midazolam_, as shown in Appendix A. This table also shows that SNPs of P-gp was not associated with AUC_fexofenadine_. This means that there is no concordance between genotype and phenotype in our study for CYP3A and P-gp.

Similar to previous models for phenotype activity predicted by the cocktail approach, CrCl and dose were found to be associated with apixaban PK parameters and only with dose for rivaroxaban. However, age and weight were also found to be significantly associated with rivaroxaban AUC_0-6h_ and C_2h_.

## 4. Discussion

Our study found that P-gp phenotypic activity impacts apixaban and rivaroxaban exposure. To the best of our knowledge, it is the first time that a metric of P-gp phenotypic activity has been linked to apixaban and rivaroxaban concentrations in vivo. These results support the important role played by P-gp in the PK process of these two drugs in vivo [11,18]. P-gp is an efflux transporter that pumps the absorbed drugs from inside the enterocyte back into the intestinal lumen, decreasing the net gut absorption [48,49]. Despite the ability of P-gp to cause the elimination of apixaban and rivaroxaban into the gut lumen, recently published in silico studies suggested that the intestinal P-gp is not clinically significant in the absorption process of apixaban and rivaroxaban [49,50]. However, these studies have some limitations, such as using mean concentration–time curves rather than individual PK profiles and PK curves with possibly insufficient time points during the absorption phase [49,50]. Moreover, a recent in vitro study used human renal cells to provide data on P-gp inhibition and showed that P-gp had a main role in the efflux of apixaban and rivaroxaban [51].

Inter-individual variability in P-gp phenotype activity can result from the influence of both exposome and/or genome [52,53,54,55,56]. Many environmental factors influence P-gp activity in hospitalized populations like ours, especially DDIs and disease state. Our results thus question whether dose adaptation should be suggested for apixaban and rivaroxaban in the presence of P-gp modulators. Currently, dose adjustment is not required when a P-gp modulator is co-administered with apixaban and rivaroxaban, but it is with edoxaban [57,58]. Indeed, edoxaban was the only DOAC for which such dose adaptation was validated in its major trial (ENGAGE AF-TIMI 48), unlike the major trials on apixaban and rivaroxaban (ARISTOTLE AF and ROCKET AF trials, respectively), which excluded potent P-gp inhibitors [59,60,61].

The present study suggests that a phenotypic switch of P-gp activity from NM to PM might result in an increase in apixaban and rivaroxaban AUC_0–6h_ by 16% and 25%, respectively [43,44]. This is far from the two- to five-fold increase in the AUC of a substrate with a specific inhibitor to allow it being classified as even a moderate inhibition [62]. However, this result is clinically relevant, as its magnitude is equivalent to a decrease in the renal function category in apixaban and an increase in the dosing regimen for rivaroxaban, according to our models. Overall, our results suggest that dose adaptation should be questioned in the presence of P-gp modulators.

Other important environmental factors impacting CYP3A/P-gp phenotype activity are the inflammation state and the renal insufficiency, particularly in hospitalized patients. Growing evidence suggests that these alter CYP/P-gp activity through cytokines and uremic toxins, respectively [53,54,63,64,65,66,67]. This may have been a confounding factor that led to the loss of significance of the effect of modulation of CYP3A on apixaban and rivaroxaban AUC_0–6h_ in our model. The effect of inflammation and renal insufficiency on P-gp activity has received less attention, but some evidence suggests an alteration of the P-gp depending on intensity, time, and isoform considered [53,68,69,70].

The absence of association between CYP3A phenotypic activity and the apixaban and rivaroxaban exposure could result from different and complementary scenarios. As CYP3A is responsible for only 15% and 18% of the metabolism of apixaban and rivaroxaban, respectively, modulation of this pathway could, therefore, have a small impact [11]. Moreover, many CYP3A modulators are also P-gp modulators, and pure CYP3A4/5 modulation may only have a modest effect on apixaban and rivaroxaban metabolism [52]. This highlights the need to further investigate the clinical relevance of combined versus single CYP3A4/P-gp modifier interactions, as studies that distinguish the relative contribution of P-gp as compared to CYP3A modulation for each known DDI are lacking [22,52,71]. This is exemplified by the fact that FDA guidelines propose to assess drug transporters modulation only if this drug clinically modulates CYP3A [71]. Consequently, PK studies and SmPCs of apixaban and rivaroxaban mention almost every time the combined effect of P-gp and CYP3A modulators and not each enzyme separately [11,14,18,19,20]. For instance, the drugs established to be responsible for thrombotic and bleeding events are mostly classified as CYP3A4/P-gp inducers and inhibitors, respectively [22,72]. However, P-gp induction is limited to human in vitro data, resulting in the assumption that the DDIs are solely attributed to a CYP3A induction [22]. Authors of a recent systematic review suggest that it is the combination of CYP3A4/P-gp inhibitors that lead to bleeding events, while a single CYP3A4 inducer or the combination of CYP3A4/P-gp inducers were responsible for thrombotic events [22]. This is in line with our results, where the impact of P-gp inhibition was more potent in increasing apixaban and rivaroxaban blood concentrations than inhibition of CYP3A alone. In addition, we observed almost no induction of CYP3A activity in our study, and this may have weakened the significance of the impact of CYP3A alone. However, the absence of association between the CYP3A phenotype activity and the PK of apixaban and rivaroxaban does not come from the absence of sufficient inter-individual variability in CYP3A phenotype activity. Indeed, the calculated coefficients of variation of CYP3A activity are 108.1% and 100.0% for the apixaban and rivaroxaban cohorts, respectively, which ensures a high inter-individual variability. The calculated coefficients of variation of P-gp activity are smaller, with 67.2% for the apixaban and 71.5% for the rivaroxaban cohorts. It is indeed known that CYP3A carries a five-fold constitutive variability due to its sensitivity to multiple factors [73]. Moreover, not all CYP3A substrates share the same specificity, and it cannot be excluded that midazolam is not a good surrogate for DOACs [74].

Overall, PK profiles appear to be significantly impacted when multi-target inhibitors are administered. Apixaban and rivaroxaban are also substrates of Breast Cancer Resistance Protein (BCRP), encoded by the *ABCG2* gene, which is another efflux transporter [23]. Two studies observed that the SNPs rs2231142 of *ABCG2* (c.421C > A) had an impact on apixaban exposure, and one case report showed a highly increased concentration of apixaban, along with other mutations on *ABCB1* and *CYP3A5* gene [28,75,76]. However, this ABCG2 efflux transporter is present in the intestine and does not seem to have a significant impact on absorption of apixaban [77]. Concerning rivaroxaban, the c.421C > A *ABCG2* mutation does not seems to have any impact, while some potential BCRP inhibitors showed an interaction with rivaroxaban [78,79]. Nevertheless, these potential BCRP inhibitors are also CYP3A and/or P-gp inhibitors. In the future, it could be interesting to measure the BCRP expression or to assess its genotype.

Another CYP is involved in the metabolism of apixaban and rivaroxaban, named CYP2J2 [23,78]. It contributes to 14% of the total clearance of rivaroxaban, which is approximately the same as the CYP3A [11]. The catalytic efficiency of CYP2J2 was assessed to be higher than that of CYP3A4 in vitro, giving a new insight of DDIs involving rivaroxaban [80]. CYP2J2*7 did not significantly impact the exposure to rivaroxaban, as observed in a study [78]. Other genetic polymorphisms of CYP2J2 have been identified, but their clinical implications are to date unknown [80,81]. For instance, ketoconazole and ritonavir have been reported to increase plasma concentration of rivaroxaban [79]. They are potential inhibitors of CYP2J2 but also strong CYP3A and P-gp/BCRP inhibitors. Therefore, it could also be valuable to measure the CYP2J2 activity in further studies assessing rivaroxaban exposure. Concerning apixaban, CYP2J2 and CYP1A2 contribute together to only 6% of the metabolism, and a significant impact of CYP2J2 is not expected [11]. Indeed, no study was found in the literature on the impact of CYP2J2 polymorphism on apixaban.

We found that renal insufficiency significantly increased the exposure to apixaban but not to rivaroxaban. This was unexpected, because all DOACs are eliminated by the kidneys [82]. It is thus largely accepted that impaired renal function directly influences the anticoagulation regimen [82]. Especially as rivaroxaban should logically be more impacted by renal dysfunction than apixaban, being 66% excreted by the kidney as compared to apixaban, which is only excreted at 25–30% [82,83].

One possible explanation for this finding is that normal and moderate renal functions were equally distributed in the apixaban cohort, whereas there were twice as many patients with normal renal function in the rivaroxaban cohort (Table 1).

Another explanation is a possibly inappropriate dosing regimen of apixaban and rivaroxaban according to creatinine clearance [84]. Rivaroxaban dose adjustment appeared to be more appropriate than apixaban dose adjustment according to renal function in our cohort, erasing the impact of this covariable. As shown in Table 1, adequate dosages (according to SmPC) were found to be prescribed in 70% of apixaban and 80% of rivaroxaban patients. This is consistent with existing real-world data that report off-label dosing of DOACs ranging from 13% to 57% [84,85]. Some studies published in the literature found that it was apixaban that had the highest rate of inappropriate dosing and others that it was rivaroxaban [84,85,86]. These rates seem dependent on the included population characteristics and could vary between studies [85]. In our study, we found a higher rate of inadequate dose selection with apixaban, which might be explained by the implication of other factors than renal function, such as age and weight [84]. Moreover, our consideration of both AF and venous thromboembolism (VTE), despite significant differences in terms of indication and dosing for VTE treatment, could be another explanation [84].

The multivariable linear regression models showed no effect of age, weight, gender, and ALAT on apixaban and rivaroxaban exposure when the phenotypic activity of CYP3A and P-gp were considered. In accordance with SmPC, gender is not a criterion for dose adjustment [8,9]. Moreover, recent population-based cohort studies did not find sex-related differences in terms of ADR occurrence or trough concentration levels of DOACs [87,88]. However, a prospective study in the perioperative setting found that female gender was a predictor for higher apixaban and rivaroxaban levels, with authors suggesting the cause being that females are at higher risk of renal insufficiency [89].

We observed an absence of association between age and apixaban PK parameters, even though it is a criterion for dose adjustment according to the SmPC [8,9]. However, our multivariable linear model found a tendency towards an increased apixaban exposure with an increase in age. The age effect could have been blurred, because most of our patients received an adequate dosage adjustment. Regarding the weight, only a small percentage of patients (16.46%) had a weight < 60 kg in our cohort, lowering the chance of finding a significant effect of this cofactor on exposure. In addition, weight and age have a small impact on exposure on their own. Indeed, subgroup analysis of the ARISTOTLE trial suggested that the presence of only one dose reduction criterion does not significantly impact the safety or efficacy [90]. Moreover, studies showed that patients with age > 65 years or weight < 50 kg were more exposed to apixaban, but there was no meaningful difference in clinical outcomes that would require dose modification based on age and weight alone [91,92]. In addition, a large register-based cohort study corroborated our results with no clear correlations found between BMI, age, and gender and trough concentrations of apixaban and rivaroxaban [88].

Phenotyping presents the advantage of measuring the effect of non-genetic factors, as compared to genotyping [93]. Indeed, even if the association between the SNPs and the enzyme activity is known, epigenetic and/or other factors can induce a phenoconversion [94,95,96]. Genotyping presents other limitations compared to phenotyping, such as the fact that functional consequences of most genetic polymorphisms have not yet been identified and that unknown/new SNPs cannot be tested [94,97,98]. Inter-individual variability in the PK of drugs due to genetic polymorphism has been identified, but our current knowledge does not allow any consistent predictions regarding patients’ drug response [55,99]. Moreover, the influence of rare *ABCB1* variants on drug bioavailability and response has not been identified yet [55]. We tested the three most prevalent SNPs, but growing evidence suggests that rare variants might have greater effects on drug PK or PD than the more common ones [55]. Unexpectedly, we found that the CYP3A predicted activity from genotype was significantly associated with the AUC_0–6h_ of rivaroxaban. However, as we found that the phenotype was not correlated to the genotype in our study, the potential physiological meaning of this association remains unexplained. This underlined the difficulty of predicting the CYP3A phenotype from the genotype [96]. Each individual genetic factor associated to CYP3A expression has a minor role, because it is regulated by multiple genes, as suggested by its known continuous and unimodal distribution [73,81]. It is, therefore, expected to not find a strong association between CYP3A genetic polymorphisms and drug exposure. A phenoconversion may have occurred in IM patients due to various environmental factors. Indeed, more than three-quarters of included patients have an IM predicted phenotype from genotype in our cohort. The high proportion of IM could be explained by the fact that the CYP3A predicted phenotype from genotype considers both the CYP3A4 and the CYP3A5. Indeed, the CYP3A5*3 has a reduced activity and is present in 95% of the European and 62% of the whole population [47]. Therefore, the probability of having an intermediate CYP3A activity is high.

Overall, as previously shown, our results highlight the need to complete genotyping by phenotyping [96]. Indeed, it was shown that performing both tests simultaneously explained more clinical events than each of the tests being performed separately [96,98].

Our study has some limitations, such as the failure to reach the target sample size, which may have resulted in a lack of power and the lack of association with CYP3A. Additionally, the protocol did not plan to carry out a full PK sampling, and certain time points could be lacking, especially in the elimination phase. Using a phenotyping cocktail approach and MR as phenotypic metrics are subject to interpretation in terms of metabolizer classifications, but these metrics were used as continuous variables to address this concern. Another limitation is the inclusion of hospitalized patients exclusively, with all the variability in non-genetic factors that this implies. This could have led to CYP3A and P-gp activity and expression being influenced by heterogenous non-genetic factors, making it difficult to extrapolate our results to other populations, such as ambulatory or non-hospitalized patients.

## 5. Conclusions

In conclusion, our results indicate that P-gp phenotypic activity, rather than P-gp polymorphisms, has a relevant impact on the exposure of apixaban and rivaroxaban. Moreover, neither CYP3A phenotypic activity nor CYP3A predicted activity from genotype had a relevant impact on the exposure of these two DOACs.

Our study suggests that integrating P-gp phenotypic activity in the dose selection criteria may be beneficial. Genotyping of *CYP3A4*, *CYP3A5*, and *ABCB1* is probably not enough to predict enzyme activity due to the dynamic application of environmental, physiological, and pathophysiological factors. More studies are needed to assess the clinical utility of adding P-gp to dose selection in terms of adverse events and efficiency.

## Figures and Tables

**Figure 1 jpm-12-00526-f001:**
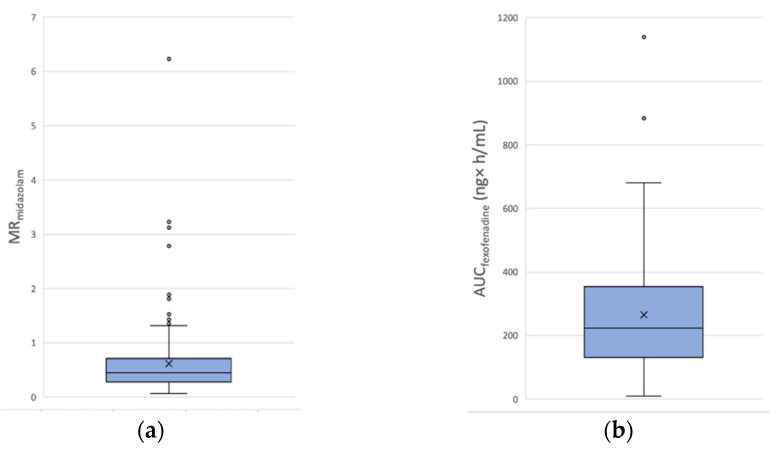
Distribution of phenotype metrics in the apixaban cohort (**a**) MR_midazolam_ and (**b**) AUC_fexofenadine_.

**Figure 2 jpm-12-00526-f002:**
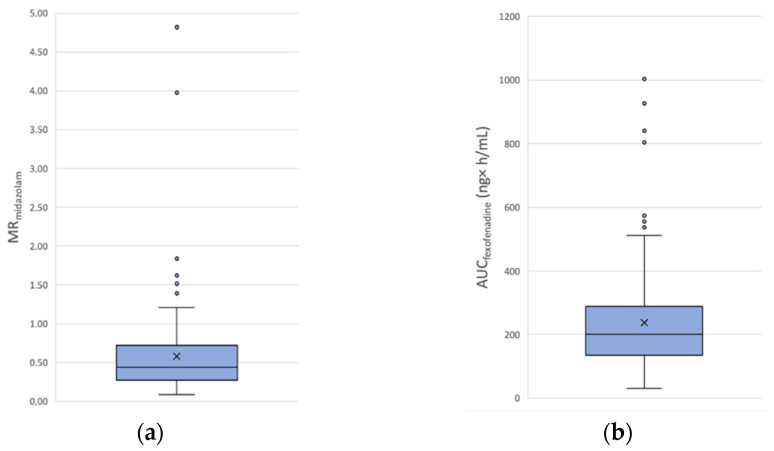
Distribution of phenotype metrics in the rivaroxaban cohort (**a**) MR_midazolam_ and (**b**) AUC_fexofenadine_.

**Figure 3 jpm-12-00526-f003:**
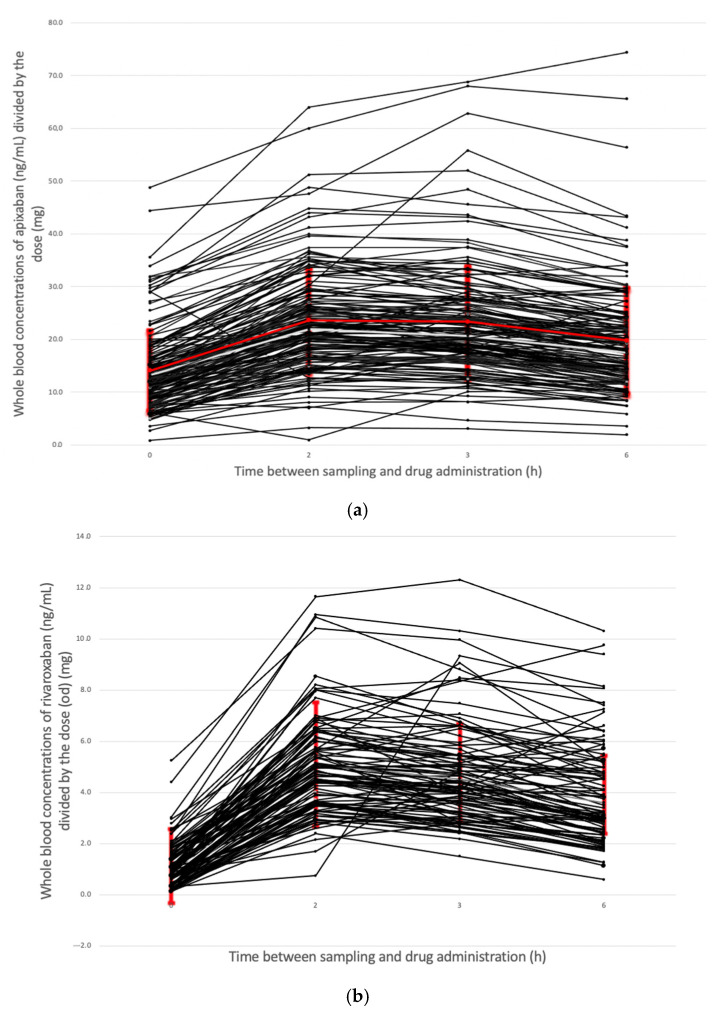
Blood concentrations normalized by the dosing regimen of (**a**) apixaban (**b**) rivaroxaban once daily and (**c**) and rivaroxaban twice daily. Each black line corresponds to an individual, and the mean ± SD is highlighted in red.

**Table 1 jpm-12-00526-t001:** Demographic characteristics of included patients with apixaban and rivaroxaban treatments.

Characteristics	Apixaban	Rivaroxaban
Gender, *n* (%)		
Male	101 (61.6%)	89 (65.9%)
Female	63 (38.4%)	46 (34.1%)
Age, mean (SD)		
Age (years)	77.4 (9.8)	71.1 (12.1)
Weight, mean (SD)		
Weight (kg)	77.4 (9.8)	82.1 (18.0)
BMI (kg/m^2^)	26.8 (5.6)	27.7 (5.7)
Indication, *n* (%)		
AF	145 (88.4%)	92 (67.6%)
DVT and PE treatment	13 (7.9%)	35 (25.7%)
DVT and PE prophylaxis	6 (3.7%)	9 (6.6%)
Doses, *n* (%)		
2.5 mg bid	70 (42.7%)	NA
5 mg bid	87 (53%)	NA
10 mg bid	7 (4.3%)	NA
10 mg od	NA	6 (4.4%)
15 mg od	NA	17 (12.6%)
20 mg od	NA	80 (59.3%)
15 mg bid	NA	32 (23.7%)
Dosage adequacy, *n* (%)		
Adequate dosage	166 (70.7%)	108 (80%)
Inadequate dosage	47 (28.7%)	27 (20%)
Unknown	1	0
Inadequate dosage, *n* (%)		
2.5 mg bid	45 (95.7%)	NA
10 mg od	NA	3 (11.1%)
15 mg od	NA	9 (33.3%)
20 mg od	NA	9 (33.3%)
15 mg bid	NA	6 (22.2%)
AF	43 (91.5%)	22 (81.5%)
Liver injury, *n* (%) or mean (SD)		
ALAT	34.5 (46.7)	38.9 (41.6)
No	149 (92.5%)	122 (90.4%)
Yes	12 (7.5%)	13 (9.6%)
Missing data	3	0
Renal function, *n* (%) or mean (SD)		
Creatinine (μ/moL)	110.8 (111.4)	95.8 (88.3)
CrCl (mL/min/1.73 m^2^) according to Cockcroft	63.8 (27.5)	77.7 (28.0)
Normal	75 (46.0%)	91 (67.4%)
Moderate	79 (48.5%)	43 (31.9%)
Severe	9 (5.5%)	1 (0.7%)
Missing data	1	0

Abbreviations: SD, standard deviation; BMI, body mass index; AF, atrial fibrillation; DVT, deep vein thrombosis; PE, pulmonary embolism; bid, twice daily; od, once daily; NA, not applicable; ALAT, alanine transaminase; CrCl, creatinine clearance.

**Table 2 jpm-12-00526-t002:** Genotype analysis results.

Characteristics	Apixaban	Rivaroxaban
Predicted phenotype from genotype CYP3A, *n* (%)		
PM	18 (11.3%)	13 (10.1%)
IM	127 (79.9%)	99 (76.7%)
NM	14 (8.8%)	17 (13.2%)
Missing data	5	6
Genotype *ABCB1* 1236C>T, *n* (%)		
No T	52 (32.1%)	44 (33.6%)
One T	74 (45.7%)	62 (47.3%)
Two T	36 (22.2%)	25 (19.1%)
Missing data	2	4
Genotype *ABCB1* 2677G>T, *n* (%)		
No T	49 (30.4%)	43 (33.1%)
One T	75 (46.6%)	56 (43.1%)
Two T	37 (23.0%)	31 (23.8%)
Missing data	3	5
Genotype *ABCB1* 3435C>T, *n* (%)		
No T	41 (25.5%)	36 (27.5%)
One T	74 (46.0%)	58 (44.3%)
Two T	46 (28.6%)	37 (28.2%)
Missing data	3	4

Abbreviations: PM, poor metabolizer; IM, intermediate metabolizer; NM, normal metabolizer; T, mutant allele.

**Table 3 jpm-12-00526-t003:** Multivariable linear regression models to assess if the phenotypic activity of CYP3A and P-gp are associated with the AUC_0–6h_ of apixaban and rivaroxaban. Each independent variable is reported with its beta coefficient (β) and its 95% confidence interval (CI95%).

	AUC_0–6h_ of Apixaban	AUC_0–6h_ of Rivaroxaban
Intercept	−46.30 (−339.86 to 247.26); *p* = 0.7557	−418.12 (−776.92 to −59.32); *p* = 0.0228
Variables		
MR_midazolam_, per log_10_	10.03 (−64.67 to 84.72); *p* = 0.7912	−90.27 (−209.99 to 29.45); *p* = 0.1381
AUC_fexofenadine_, per log_10_	173.96 (77.33 to 270.58); ***p* = 0.0005**	232.51 (105.69 to 359.33); ***p* = 0.0004**
Weight, per kg	−0.25 (−2.12 to 1.61); *p* = 0.7881	1.19 (−0.63 to 3.01); *p* = 0.1973
CrCl, per unit	−2.13 (−3.72 to −0.54); ***p* = 0.0091**	−0.44 (−2.53 to 1.65); *p* = 0.6675
ALAT, per unit	0.44 (−0.91 to 1.79); *p* = 0.5217	0.08 (−0.65 to 0.81); *p* = 0.8314
Gender		
Male	Reference category	Reference category
Female	43.31 (−15.31 to 101.92); *p* = 0.1464	44.63 (−27.63 to 116.89); *p* = 0.2238
Dose		
2.5 mg bid	Reference category; ***p* < 0.0001 ***	NA
5 mg bid	279.44 (221.14 to 337.75); *p* < 0.0001	NA
10 mg bid	688.81 (478.01 to 889.60); *p* < 0.0001	NA
10 mg od	NA	Reference category; ***p* = 0.0045 ***
15 mg od	NA	90.97 (−32.04 to 213.99); *p* = 0.1457
20 mg od	NA	165.18 (37.54 to 292.83); *p* = 0.0116
15 mg bid	NA	180.44 (64.84 to 296.02); *p* = 0.0025
Age		
<65 years	Reference category; *p* = 0.2064 *	Reference category; *p* = 0.1191 *
65–74 years	95.82 (−10.13 to 201.76); *p* = 0.0759	81.34 (15.35 to 147.32); *p* = 0.0161
75–84 years	82.33 (−22.35 to 187.01); *p* = 0.1222	59.56 (−53.36 to 172.48); *p* = 0.2984
>85 years	113.34 (−1.13 to 227.82); *p* = 0.0523	56.06 (−54.25 to 166.38); *p* = 0.3163

* *p*-value for the overall association between AUC_0–6h_ and the variable. Abbreviations: CrCl, creatinine clearance; ALAT, alanine transaminase; AUC, area under the curve; bid, twice daily; od, once daily; NA, not applicable. Statistically significant values are marked with bold.

**Table 4 jpm-12-00526-t004:** Multivariable linear regression models to assess whether the genotype of CYP3A and P-gp are associated with the AUC_0–6h_ of apixaban and rivaroxaban. Each independent variable is reported with its beta coefficient (β) and its 95% confidence interval (CI95%).

	AUC_0–6h_ of Apixaban	AUC_0–6h_ of Rivaroxaban
Intercept	398.75 (196.35 to 601.15); *p* = 0.0002	−39.45 (−424.81 to 345.91); *p* = 0.8396
Variables		
Weight, per kg	−0.11 (−2.08 to 1.86); *p* = 0.9142	1.89 (0.05 to 3.72); ***p* = 0.0442**
CrCl, per unit	2.38 (−3.97 to −0.79); ***p* = 0.0036**	−0.43 (−2.90 to 2.05); *p* = 0.7333
ALAT, per unit	0.45 (−1.20 to 2.11); *p* = 0.5893	0.34 (−0.35 to 1.04); *p* = 0.3280
Predicted phenotype from genotype CYP3A		
IM	Reference category; *p* = 0.2154 *	Reference category; ***p* = 0.0021 ***
NM	−15.50 (−139.46 to 108.47); *p* = 0.8051	142.49 (56.08 to 228.90); *p* = 0.0014
PM	−77.10 (-164.78 to 10.58); *p* = 0.0843	134.23 (−24.82 to 293.28); *p* = 0.0973
Genotype *ABCB1* 1236C>T		
No mutation	Reference category; *p* = 0.9723 *	Reference category; *p* = 0.4955 *
Heterozygous for mutation	−12.33 (−138.96 to 114.30); *p* = 0.8476	−46.50 (−163.59 to 70.59); *p* = 0.4329
Homozygous for mutation	−20.09 (−190.52 to 150.34); *p* = 0.8160	21.46 (−125.94 to 168.86); *p* = 0.7735
Genotype *ABCB1* 3435C>T		
No mutation	Reference category; *p* = 0.5600 *	Reference category; *p* = 0.2663 *
Heterozygous for mutation	−51.58 (−149.70 to 46.54); *p* = 0.3004	−51.69 (−170.92 to 67.54); *p* = 0.3921
Homozygous for mutation	−18.22 (−112.45 to 76.01); *p* = 0.7028	−71.90 (−161.27 to 17.46); *p* = 0.1137
Genotype *ABCB1* 2677G>T		
No mutation	Reference category; *p* = 0.9069 *	Reference category; *p* = 0.6892 *
Heterozygous for mutation	29.83 (−106.91 to 166.57); *p* = 0.6669	56.52 (−75.24 to 188.29); *p* = 0.3971
Homozygous for mutation	32.15 (−138.11 to 202.40); *p* = 0.7095	54.86 (−96.09 to 205.81); *p* = 0.4728
Gender		
Male	Reference category	Reference category
Female	50.24 (−19.91 to 120.38); *p* = 0.3004	46.71 (−29.08 to 122.49); *p* = 0.2246
Dose		
2.5 mg bid	Reference category; ***p* < 0.0001 ***	NA
5 mg bid	275.77 (201.67 to 349.87); *p* < 0.0001	NA
10 mg bid	689.14 (470.72 to 907.55); *p* < 0.0001	NA
10 mg od	NA	Reference category; ***p* = 0.0012 ***
15 mg od	NA	128.66 (−17.23 to 274.55); *p* = 0.0833
20 mg od	NA	250.76 (117.30 to 384.22); *p* = 0.0003
15 mg bid	NA	221.57 (75.08 to 368.07); *p* = 0.0034
Age		
<65 years	Reference category; *p* = 0.3989 *	Reference category; ***p* = 0.0062 ***
65–74 years	81.29 (−37.74 to 200.33); *p* = 0.1719	138.27 (61.87 to 214.67); *p* = 0.0005
75–84 years	57.29 (−50.00 to 164.58); *p* = 0.2929	127.49 (−17.50 to 272.48); *p* = 0.0842
>85 years	104.66 (−32.37 to 241.70); *p* = 0.1333	124.29 (−14.68 to 263.25); *p* = 0.0791

* *p*-value for the overall association between AUC_0–6h_ and the variable. Abbreviations: CrCl, creatinine clearance; ALAT, alanine transaminase; IM, intermediate metabolizer; NM, normal metabolizer; PM, poor metabolizer; bid, twice daily; od, once daily; NA, not applicable. Statistically significant values are marked with bold.

## Data Availability

The data presented in this study are available on request from the corresponding author.

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
