# Peer review of "Impact of the Genotype and Phenotype of CYP3A and P-gp on the Apixaban and Rivaroxaban Exposure in a Real-World Setting"

_jpm, 2022, doi:10.3390/jpm12040526_

Round 1

Reviewer 1 Report

This is an extremely important and valuable paper with robust methodology, good size of the patient groups, in addition, it is clinically important and very interesting to read! The "Geneva Cocktail" - is a nice method to measure CYP3A4/5 phenotypic activity!

Author Response

Point 1: This is an extremely important and valuable paper with robust methodology, good size of the patient groups, in addition, it is clinically important and very interesting to read! The "Geneva Cocktail" - is a nice method to measure CYP3A4/5 phenotypic activity!

Response 1: We thank the reviewer for his appreciation.

Reviewer 2 Report

This article entitled “Impact of the Genotype and Phenotype of CYP3A and P-gp on the Apixaban and Rivaroxaban Exposure in a Real-World Setting” is well organized. The authors investigated the relations between the exposure of apixaban and rivaroxaban and P-gp/CYP3A by a prospective study conducted among 300 patients. In particular, the authors distinguished the genotype and phenotype. Data analysis showed the P-gp activity and GFR significantly affected the apixaban PK process, and P-gp activity also influenced the rivaroxaban exposure. While the genotype and phenotype of CYP3A were not relevant to the two DOACs exposure. The study will be helpful to medication adaption in clinic, but some problems still exist in the manuscript.

Here is my major concern:

  1. Multiple clinical trials of rivaroxaban pharmacokinetics in human showed that single-target inhibitors of CYP3A4 or P-gp did not affect rivaroxaban in vivo. When rivaroxaban was co-administered with multitarget inhibitors, the clinically significant change happened in rivaroxaban pharmacokinetics. It indicated that multiple targets involved in the rivaroxaban pharmacokinetic process and a single factor may have little effect. The viewpoint has also been mentioned in the manuscript. However, rivaroxaban is also the substrate drug to BCRP [Br J Clin Pharmacol. 2013 Sep;76(3):455-66. doi: 10.1111/bcp.12075.]. Additionally, BCRP was mainly distributed in the intestine and the kidney, which overlapped with rivaroxaban absorption and excretion. Thus, it is suggested to add the investigation or discussion on BCRP in the manuscript.
  2. In addition to CYP3A4, CYP2J2 also participated in the process of rivaroxaban in vivo. Notably, the catalytic efficiency of CYP2J2in vitro was similar to that of CYP3A4[Cardiovasc Drugs Ther. 2022 Feb;36(1):121-129. doi: 10.1007/s10557-020-07129-z. Epub 2021 Jan 7.], and some SNPs of CYP2J2 could regulate the CYP2J2 activity [Pharmacol Ther. 2013 Apr;138(1):103-41. doi: 10.1016/j.pharmthera.2012.12.007. Epub 2013 Jan 16.]. Additionally, ketoconazole and ritonavir, both of them significantly influence rivaroxaban pharmacokinetics in vivo, also potently inhibited CYP2J2[Drug Metab Dispos. 2016 Mar;44(3):453-9. doi: 10.1124/dmd.115.067744. Epub 2015 Dec 14; Drug Metab Dispos. 2016 Mar;44(3):453-9. doi: 10.1124/dmd.115.067744. Epub 2015 Dec 14; Drug Metab Dispos. 2012 May;40(5):943-51. doi: 10.1124/dmd.111.043505. Epub 2012 Feb 10.]. So, the role of CYP2J2 in rivaroxaban pharmacokinetic process is suggested to add to the discussion.
  3. Authors investigated the effect of CYP3A genotype polymorphism on the two DOACs. Drug oxidation phenotypes of CYP3A4 are strongly variable, but unimodally distributed and most SNPs are not related to activity regulation [Pharmacol Ther. 2013 Apr;138(1):103-41. doi: 10.1016/j.pharmthera.2012.12.007. Epub 2013 Jan 16.]. Additionally, although some SNPs of CYP3A5 may regulate enzyme activity, CYP3A5 contributed little to the two drugs' metabolism. Therefore, exploring the relationship between the genotype and phenotype of CYP3A4, CYP3A5 and the two drugs may be lack clinical significance.

Here are some minor matters:

  1. The description of results in the abstract was in a confusing order. It is suggested to describe the positive results first (P-gp/GFR) and then state the negative results (CYP3A).
  2. There are multiple mistakes or typo errors in writing in the manuscript, for example, an LC-MS (line 186), was administrated (line 189), a higher risk (line 448), came from (line 450), could be explained (452). Please check and revise.
  3. Some tense problems exist in the manuscript, please check and revise.

Author Response

This article entitled “Impact of the Genotype and Phenotype of CYP3A and P-gp on the Apixaban and Rivaroxaban Exposure in a Real-World Setting” is well organized. The authors investigated the relations between the exposure of apixaban and rivaroxaban and P-gp/CYP3A by a prospective study conducted among 300 patients. In particular, the authors distinguished the genotype and phenotype. Data analysis showed the P-gp activity and GFR significantly affected the apixaban PK process, and P-gp activity also influenced the rivaroxaban exposure. While the genotype and phenotype of CYP3A were not relevant to the two DOACs exposure. The study will be helpful to medication adaption in clinic, but some problems still exist in the manuscript.

Here is my major concern:

Point 1: Multiple clinical trials of rivaroxaban pharmacokinetics in human showed that single-target inhibitors of CYP3A4 or P-gp did not affect rivaroxaban in vivo. When rivaroxaban was co-administered with multitarget inhibitors, the clinically significant change happened in rivaroxaban pharmacokinetics. It indicated that multiple targets involved in the rivaroxaban pharmacokinetic process and a single factor may have little effect. The viewpoint has also been mentioned in the manuscript. However, rivaroxaban is also the substrate drug to BCRP [Br J Clin Pharmacol. 2013 Sep;76(3):455-66. doi: 10.1111/bcp.12075.]. Additionally, BCRP was mainly distributed in the intestine and the kidney, which overlapped with rivaroxaban absorption and excretion. Thus, it is suggested to add the investigation or discussion on BCRP in the manuscript.

Response 1: We thank the reviewer for this valuable comment. We do not have the possibility to do these investigations on the BCRP transporter, but we have added this notion to the discussion as suggested (line 471-482).

Point 2: In addition to CYP3A4, CYP2J2 also participated in the process of rivaroxaban in vivo. Notably, the catalytic efficiency of CYP2J2in vitro was similar to that of CYP3A4[Cardiovasc Drugs Ther. 2022 Feb;36(1):121-129. doi: 10.1007/s10557-020-07129-z. Epub 2021 Jan 7.], and some SNPs of CYP2J2 could regulate the CYP2J2 activity [Pharmacol Ther. 2013 Apr;138(1):103-41. doi: 10.1016/j.pharmthera.2012.12.007. Epub 2013 Jan 16.]. Additionally, ketoconazole and ritonavir, both of them significantly influence rivaroxaban pharmacokinetics in vivo, also potently inhibited CYP2J2[Drug Metab Dispos. 2016 Mar;44(3):453-9. doi: 10.1124/dmd.115.067744. Epub 2015 Dec 14; Drug Metab Dispos. 2016 Mar;44(3):453-9. doi: 10.1124/dmd.115.067744. Epub 2015 Dec 14; Drug Metab Dispos. 2012 May;40(5):943-51. doi: 10.1124/dmd.111.043505. Epub 2012 Feb 10.]. So, the role of CYP2J2 in rivaroxaban pharmacokinetic process is suggested to add to the discussion.

Response 2: As suggested by the reviewer, a paragraph concerning the role of CYP2J2 in rivaroxaban and apixaban pharmacokinetic processes has been added to the discussion section (line 483-495).

Point 3: Authors investigated the effect of CYP3A genotype polymorphism on the two DOACs. Drug oxidation phenotypes of CYP3A4 are strongly variable, but unimodally distributed and most SNPs are not related to activity regulation [Pharmacol Ther. 2013 Apr;138(1):103-41. doi: 10.1016/j.pharmthera.2012.12.007. Epub 2013 Jan 16.]. Additionally, although some SNPs of CYP3A5 may regulate enzyme activity, CYP3A5 contributed little to the two drugs' metabolism. Therefore, exploring the relationship between the genotype and phenotype of CYP3A4, CYP3A5 and the two drugs may be lack clinical significance.

Response 3: We thank the reviewer for his substantial comment. A commentary in that extent has been added to the discussion section (line 569 to 572).

Here are some minor matters:

Point 4: The description of results in the abstract was in a confusing order. It is suggested to describe the positive results first (P-gp/GFR) and then state the negative results (CYP3A).

Response 4: We agree with the reviewer and the abstract has been modified accordingly (line 32 to 33).

Point 5: There are multiple mistakes or typo errors in writing in the manuscript, for example, an LC-MS (line 186), was administrated (line 189), a higher risk (line 448), came from (line 450), could be explained (452). Please check and revise. Some tense problems exist in the manuscript, please check and revise.

Response 5: The whole manuscript was checked and revised. We hope to have identified all errors.

Round 2

Reviewer 2 Report

The authors have carefully answered all questions.